# Efficient Skill Discovery via Regret-Aware Optimization

He Zhang [1 2]   Ming Zhou [2]   Shaopeng Zhai [2]   Ying Sun [1]   Hui Xiong [1 3]

## Abstract

Unsupervised skill discovery aims to learn diverse and distinguishable behaviors in open-ended reinforcement learning. For existing methods, they focus on improving diversity through pure exploration, mutual information optimization, and learning temporal representation. Despite that they perform well on exploration, they remain limited in terms of efficiency, especially for the high-dimensional situations. In this work, we frame skill discovery as a min-max game of skill generation and policy learning, proposing a regret-aware method on top of temporal representation learning that expands the discovered skill space along the direction of upgradable policy strength. The key insight behind the proposed method is that the skill discovery is adversarial to the policy learning, i.e., skills with weak strength should be further explored while less exploration for the skills with converged strength. As an implementation, we score the degree of strength convergence with regret, and guide the skill discovery with a learnable skill generator. To avoid degeneration, skill generation comes from an up-gradable population of skill generators. We conduct experiments on environments with varying complexities and dimension sizes. Empirical results show that our method outperforms baselines in both efficiency and diversity. Moreover, our method achieves a 15% zero shot improvement in high-dimensional environments, compared to existing methods.

[1]Thrust of Artificial Intelligence, The Hong Kong University of Science and Technology (Guangzhou) [2]Shanghai AI Lab [3]Department of Computer Science and Engineering, The Hong Kong University of Science and Technology Hong Kong SAR. Correspondence to: Ming Zhou <zhouming@pjlab.org.cn>, Ying Sun <yings@hkust-gz.edu.cn>, Hui Xiong <xionghui@ust.hk>.

*Proceedings of the 42nd International Conference on Machine Learning*, Vancouver, Canada. PMLR 267, 2025. Copyright 2025 by the author(s).

## 1. Introduction

Deep Reinforcement Learning (RL) has demonstrated its potential to surpass human decision-making abilities in both simulated environments and real-world applications, such as gaming (Silver et al., 2018; Vinyals et al., 2019), recommendation systems (Xu et al., 2019; Zheng et al., 2023), urban computing (Sun et al., 2023), and robotic control (Black et al., 2024). However, RL is still unable to self-improve autonomously by accumulating new skills without end, preventing the achievement of artificial general intelligence (Hughes et al., 2024; Haber et al., 2018; Colas et al., 2022). The principle of Unsupervised Skills Discovery (USD) aligns with this challenge, which focuses on autonomously summarizing diverse and useful skills for downstream tasks without explicit rewards.

USD has emerged from the field of unsupervised RL (Hao et al., 2023; Eysenbach et al., 2022), exploring environments without external rewards based on information theory. Pioneering work (Gregor et al., 2016; Eysenbach et al., 2019; Laskin et al., 2022) in this area introduced the idea of using diverse and distinguishable skills by calculating mutual information. Building on this, LSD (Park et al., 2022) achieved a novel discovery of temporal dynamic skills by learning temporal representations of states. Subsequently, METRA (Park et al., 2024c) combined temporal representation learning with mutual information and surprisingly demonstrated that their method is scalable, as it closely resembles PCA clustering. However, these approaches assume that the skills are uniformly distributed, which simplifies the optimization objective. Other works (Lee & Seo, 2023; Wang et al., 2024a) have explored skill selection policies within hierarchical frameworks to address changing environments for downstream tasks. However, they did not fully investigate the exploration of diverse skills within a single environment. In general, previous work has overlooked the critical role of skill selection in improving learning efficiency and enhancing the diversity of skills.

To address these challenges, our key insights are as follows: (1) skill discovery should be grounded by the strength of the policy, sharing a similar idea in auto-curriculum learning (Jiang et al., 2021b); and (2) skills may exhibit sequential dependencies, requiring some to be acquired before others. For instance, in a manipulation task, the agent should

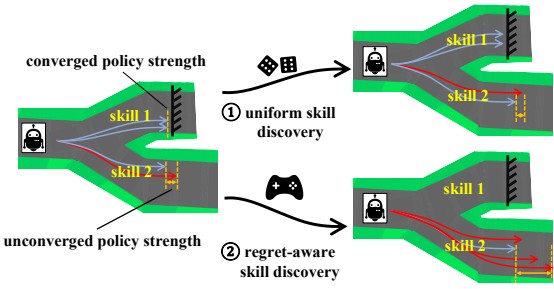

*Figure 1.* Comparison between uniform skill discovery and regret-aware skill discovery. Blue lines indicate skills with converged strength (low regret). Red lines indicate skills need further explore (high regret). By re-balancing exploration based on regret signals, the regret-aware method exhibits improved efficiency.

learn a basic skill '*picking up*' before the advanced skill '*opening the grip*' (Liu et al., 2024). Intuitively, for the skills that have been, or can no longer be further mastered, the policy strength for these skills towards converging; whereas the policy strength for the unmastered skills can be further improved. Therefore, to quantify this principle, we consider using *regret* to measure the convergence of policy strength.

In the context of USD, regret quantifies the discrepancy between the actual policy strength and the maximum strength that the agent could have achieved. Lower regret indicates faster convergence to skills, whereas higher regret suggests slower convergence. As illustrated in Figure 1, incorporating regret awareness into skill discovery can enhance learning efficiency compared to naive methods. By leveraging regret signals to rebalance exploration, policy learning prioritizes exploration of underconverged skills, rather than indiscriminate exploration of the entire skill space.

On top of these observations, we propose an efficient **R**egret-aware **S**kill **D**iscovery (RSD) framework for USD, which is framed as a min-max adversarial optimization algorithm. For implementation, RSD comprises two interleaved processes: (1) learning an agent policy within a bounded representation space to minimize regrets for generated skills; and (2) learning a population of Regret-aware Skill Generator (RSG) for skill generation towards maximizing regrets. The skill learning process builds on METRA (Park et al., 2024c), yet it struggles to distinguish interdependent or similar skills at the representation level due to ignoring the magnitude of skill vectors. As an improvement, we propose a bounded representation learning approach for agent policy, which significantly enhances the diversity of skills. Moreover, we found that using a single learnable skill generator at a time leads to skill forgetting, limiting skill diversity, hindering efficiency, and resulting in degeneration (Wang et al., 2024b), which is a common issue in previous work. To address this, we propose maintaining an updatable population of skill

generators, which ensures both the skill diversity and their learnability by the agent policy.

We evaluate our approach across environments with varying state dimensions and complex map structures. The results show that our method significantly improves learning efficiency and increases skill diversity in complex environments. In addition, our method achieves up to a 16% improvement in zero-shot evaluation compared to baselines.

## 2. Preliminary

**Conditioned MDP.** Following prior work (Park et al., 2024c), we formulate the unsupervised skill discovery problem within a conditioned Markov Decision Process (MDP) framework. Conditioned MDP is formulated as $M = \langle S, A, P, R, \mu_0, Z \rangle$. The state space is $S \subseteq \mathbb{R}^n$, with $s \in S$ denoting an individual state with $n$ dimensions. The action space is $A \subseteq \mathbb{R}^m$, where $a \in A$ represents an individual action with dimensions $m$. The transition dynamics is given by the transition function $P(s'|s, a) \in [0, 1]$. $\mu_0(s)$ denotes the initial state distribution over $S$. And $Z = \{z | z \in \mathbb{R}^d\}$ denotes the latent space of the skill. Given a skill $z$, the skill-conditioned policy (Kim et al., 2021) can be defined as $\pi(a \mid s, z) \in [0, 1]$ that satisfies $\sum_{a \in A} \pi(a \mid s, z) = 1$. With the time horizon denoted as $T$, the trajectory derived from $\pi$ can be formulated as :

$$\tau = \{(s_t, a_t, s_{t+1})\}_{t=0}^{T-1}.$$

If $p(z)$ denotes the distribution of skills, we have

$$p^\pi(\tau \mid z) = \mu_0 p(z) \prod_{t=0}^{T-1} \pi(a_t \mid s_t, z) P(s_{t+1} \mid s_t, a_t),$$

which indicates the probability of $\tau$ derived from $\pi$ and $z$. In the context of conditioned MDP, the learning objective of the unsupervised skill discovery problem is to find a policy $\pi$ that maximizes the expectation of the cumulative return over the whole skill space $Z$, which can be formulated as :

$$V_\pi(s) = \mathbb{E}_{\substack{a_t \sim \pi(\cdot | s_t, z) \\ z \sim p(z) \\ s_{t+1} \sim P(\cdot | s_t, a_t)}} \left( \sum_t \gamma^t r(s_t, s_{t+1}, z) \mid s_0 = s \right),$$

$$\tag{1}$$

where $r(s_t, s_{t+1}, z) \in R$ and $R : S \times S \times Z \to [-1, 1]$.

**Temporal Representation for Mutual Information.** Previous works (Gregor et al., 2016; Eysenbach et al., 2019; Laskin et al., 2022) achieve skill discovery by maximizing the mutual information (MI) between states and skills:

$$I(S; Z) = \mathbb{E}_{p(s,z)} \left[ \log \frac{p(s, z)}{p(s)p(z)} \right].$$

Based on the MI objective, METRA (Park et al., 2024c) leverages Wasserstein Dependency Measure (WDM) (Ozair

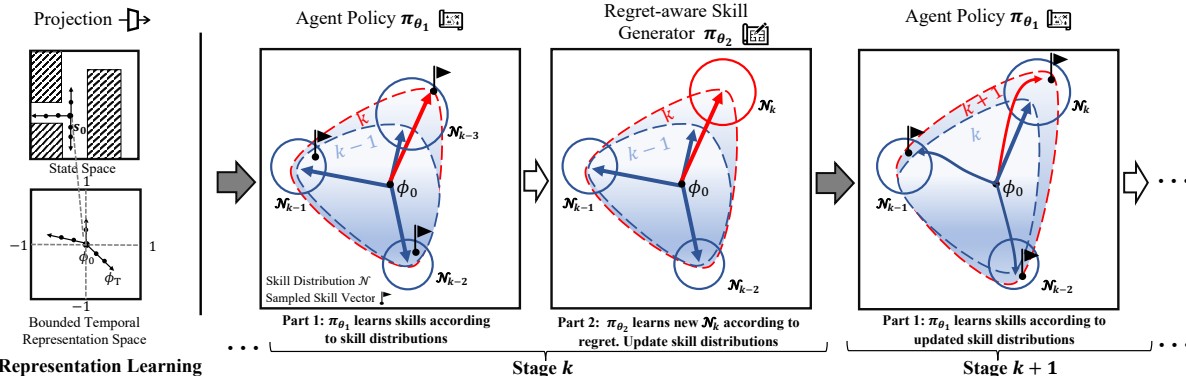

*Figure 2.* Overview of RSD. The leftmost illustrates how the state space is projected onto a bounded temporal representation space. The right side shows learning process of the skill generator: circles denotes skill distributions, dashed regions indicate coverage, the larger the more diverse, solid lines represent trajectories, and flags correspond to skills sampled from these distributions. Collectively, these circles form the population of skill generators for the agent's policy. In Part 1, the agent policy expands its coverage by mastering the skills (as seen by the red region outgrowing the blue region). In Part 2, the RSG trains a new $\pi_{\theta_2}$ to generate high-regret skills, depicted by a red circle replacing a blue one. Here, $\mathcal{N}_k$ represents the skill distribution of the current $\pi_{\theta_2}$.

et al., 2019) to actively maximize the representation distance between different skill trajectories. The objective can be expressed using Kantorovich-Rubenstein duality as follows:

$$I_{\mathcal{W}}(S; Z) = \sup_{\|f\|_L \leq 1} \mathbb{E}_{p(s,z)}[f(s, z)] - \mathbb{E}_{p(s)p(z)}[f(s, z)], \tag{2}$$

where $f$ scores the similarity between each $s$ and $z$, and is formulated as an inner production of the state representation $\phi(s)$ and $z$, i.e., $f(s, z) = \phi(s_t)^\top z$. With further derivation, Eq. (2) is equivalent to maximize the following objective:

$$I_{\mathcal{W}}(s_T; Z) \approx \sup_{\|\phi\|_L \leq 1} \mathbb{E}_{p(\tau, z)} \left[ \sum_{t=0}^{T-1} (\phi(s_{t+1}) - \phi(s_t))^\top z \right],$$
$$\text{s.t.} \quad \sum_{t=0}^{h-1} \|\phi(s_{t+1}) - \phi(s_t)\| \leq 1. \tag{3}$$

Thus, maximization of $I_{\mathcal{W}}$ can be achieved by learning a temporal representation $\phi(\cdot)$. With that, METRA further gives a unified reward function for policy learning, i.e.,

$$r(s_t, s_{t+1}, z) = (\phi(s_{t+1}) - \phi(s_t))^\top z. \tag{4}$$

The RL process for the agent policy of our method builds upon METRA, while incorporating additional enhancements in both representation learning and reward function design. These improvements ensure skill differentiation and promote diversity, thus enhancing learning efficiency.

## 3. Methodology

Our approach learns diverse and competent behaviors via regret-aware skill discovery. At a high level, RSD can be

viewed as a min-max optimization between these two processes: (1) training a **agent policy** $\pi_{\theta_1}$ to minimize the regret incurred under a set of skills and (2) optimizing a **skill generator policy** $\pi_{\theta_2}$ to maximize this regret by proposing new challenging skills. We summarize the overall framework in Figure 2 and present the complete procedure in Algorithm 1. The functionalities of the two components are described:

- **Agent Policy** $\pi_{\theta_1}$: This policy maps a state and a skill vector $z$ to an action. At each iteration, we sample a skill $z$ from a population of skill generators, and condition the agent policy on $z$ to perform reinforcement learning. The objective is to improve policy competence and reduce regret in the current skill space.

- **Skill Generator Policy** $\pi_{\theta_2}$: In contrast, this policy operates entirely in the latent skill space and generates new skill vectors. Given a fixed agent policy $\pi_{\theta_1}$, we optimize $\pi_{\theta_2}$ to generate skills that induce high regret for the agent. After optimization, $\pi_{\theta_2}$ is added to the population of the skill generator $P_z$, while an outdated generator is removed to maintain population diversity.

In the following sections, we first formalize the min-max optimization framework to claim the global optimization objective, then describe the training of $\pi_{\theta_1}$, and present the regret-maximizing optimization of the skill generator $\pi_{\theta_2}$.

### 3.1. Min-max Optimization for USD

In this section, we formulate our framework as a min-max adversarial optimization for USD. We begin by separately analyzing the optimization objectives of the two consecutive processes both from the perspective of regret, and then unify them to represent the overall optimization objective.

**Algorithm 1** RSD

---

1: Initialize $\pi_{\theta_1}, \pi_{\theta_2}, P_z, \phi, \lambda$
2: **for** $k = 1, 2, \ldots, $ Epoch_max **do**
3:     **Part 1: Updating the $\pi_{\theta_1}$**
4:     Sample a batch of $z$ from $P_z$
5:     Collect trajectories using $\pi_{\theta_1}(\cdot|z)$
6:     **for** $j = 1, 2, \ldots, $ Agent_Policy_Training_Steps **do**
7:         Sample transitions from buffer: $(s_t, s_{t+1}, a, z)$
8:         Update $\phi$ to maximize $I_\phi(s_t, s_{t+1}, z) + \lambda C_\phi(s_t, s_{t+1})$
9:         Update $\lambda$ to minimize $\lambda C_\phi(s_t, s_{t+1})$
10:        Update $\pi_{\theta_1}$ using SAC with reward $r_\phi(s_t, s_{t+1}, z)$
11:     **end for**
12:     **Part 2: Updating the $\pi_{\theta_2}$**
13:     **for** $i = 1, 2, \ldots, $ RSG_Training_Steps **do**
14:         Sample $z$ from initialized $\pi_{\theta_2}$
15:         Update $\pi_{\theta_2}$ to maximize $J_{\theta_2}(z)$ according to $\pi_{\theta_1}^k, \pi_{\theta_1}^{k-1}$
16:     **end for**
17:     Update skill population $P_z$ using $\pi_{\theta_2}$
18: **end for**

---

The RL process for $\pi_{\theta_1}$ aims to maximize the cumulative intrinsic return related to skills, according to Eq. 4. According to ideas from no-regret algorithms (Sutton, 2018), the value of regret reflects the policy's learning strength. Thus, in the context of USD, regret conditioned on $z$ serves as the discrepancy between the actual strength of the skill policy and the maximum strength that the agent could potentially have achieved. However, calculating regret based on its original definition assumes that the optimal cumulative return from the optimal policy can be determined, which is challenging under our circumstances. Inspired by previous work (Rutherford et al., 2024), we use the TD error in n steps to estimate the value of regret. We assume that the regret at learning stage $k$ can be calculated as follows:

$$Reg_k = V_k - V_{k-1}, \quad (5)$$

where $V$ is the value function of the RL policy as defined in Eq. 1. Consequently, the objective of the agent's policy can be regarded as minimizing regret after each iteration.

In addition, the population $P_z$ of $\pi_{\theta_2}$ aims to identify the most promising skills with a significant lack of convergence to learn. The regret, estimated using TD-error, also serves as a feasible measure for this purpose. A positive regret for a skill indicates that the latest learning policy surpasses the previous one, and the larger the regret, the more under-converged strength the skill possesses. Thus, the objective of $\pi_{\theta_2}$ can be regarded as finding a distribution that corresponds to the maximum regret at stage $k$.

To explicitly estimate the value of regret, we first calculate the value function of $\pi_{\theta_1}$, which is optimized using the

SAC (Haarnoja et al., 2018) framework. The value function conditioned on $z$ can be formulated as following equation:

$$V_\pi(s \mid z) = \mathbb{E}_{a \sim \pi}\left[Q_\pi(s, a, z) - \alpha \log \pi(a|s, z)\right], \quad (6)$$

where $Q_\pi(s, a, z)$ is the Q function, and $\alpha$ denotes the entropy regularization coefficient defined in SAC. The value function at stage $k$ is defined as $V_{\pi_{\theta_1}^k}(s_0 \mid z)$. After each stage, the agent policy moves closer to the optimal solution. Thus, the improvement of $V_{\pi_{\theta_1}^k}(s_0 \mid z)$ means that $\pi_{\theta_1}$ learns a better skill behavior starting from $s_0$. The regret of $z$ at stage $k$ can be estimated using the following equation:

$$Reg_k(z) = V_{\pi_{\theta_1}^k}(s_0 \mid z) - V_{\pi_{\theta_1}^{k-1}}(s_0 \mid z), \quad (7)$$

where $\pi_{\theta_1}^k$ represents the agent policy at learning stage $k$.

Considering these two policies as a min-max adversarial optimization RL problem (Durugkar et al., 2021), we describe our algorithm in following mathematical form:

$$\min_{\theta_1} \max_{\theta_2} \mathbb{E}_{z \sim P_z}\left[V_{\pi_{\theta_1}^k}(s_0 \mid z) - V_{\pi_{\theta_1}^{k-1}}(s_0 \mid z)\right]. \quad (8)$$

### 3.2. Agent Policy Learning in RSD

Following the formulation in Section 2, the agent's policy learning consists of representation optimization and policy optimization. According to Eq. 7, searching $z$ in an unbounded space for the maximum of regret is infeasible. To address this limitation, we introduce representation optimization within a bounded representation space. Our bounded representation function is defined as $\phi(\cdot)$ that satisfies $\|\phi\|_\infty \leq 1$ by employing the function $tanh(\cdot)$.

In order to represent more information, we assume that the skill $z$ can be any arbitrary vector in a bounded space. Due to this transformation, the scale of the inner product output changes, which undermines the learning stability of $\phi$. To address the problem, we design a $\vec{z}_{updated}$ to replace the unit $\vec{z}$, which is formulated as:

$$\vec{z}_{updated} = \frac{z - \phi(s_t)}{\|z - \phi(s_t)\|}. \quad (9)$$

In this way, the information from the magnitude of $z$ can be used and the scale of the inner product keeps stable at the same time. The objective of $\phi$ can be formulated as:

$$I_\phi(s_t, s_{t+1}, z) = \mathbb{E}_{p(\tau, z)}\left[\sum_{t=0}^{T-1}(\phi(s_{t+1}) - \phi(s_t))^\top \cdot \vec{z}_{updated}\right]. \quad (10)$$

Due to the bounded nature of $\phi(\cdot)$, it naturally satisfies the Lipschitz constraint. However, an additional constraint arises from the maximum number of timesteps, dictated by

the temporal relationship. A well-learned skill trajectory $\{(s_t, a, s_{t+1}, z)\}_{t=0}^{T-1}$ in the bounded space must hold:

$$\sum_{t=0}^{T-1} \|\phi(s_{t+1}) - \phi(s_t)\| \leq 1. \tag{11}$$

A direct way to satisfy this constraint is to set an upper bound on the distance at each timestep. Thus, the constraint on the representation function $\phi$ can be formulated as:

$$\frac{1}{T} - \|\phi(s_{t+1}) - \phi(s_t)\| \geq 0, \quad \forall t \in \{0, \ldots, T-1\}.$$

Using the dual objective to optimize the constrained optimization problem, the final objective of the representation function $\phi$ can be formulated as:

$$\min_{\lambda} \max_{\phi} \ I_\phi(s_t, s_{t+1}, z) + \lambda C_\phi(s_t, s_{t+1}), \tag{12}$$

where $C_\phi(s_t, s_{t+1}) = \min(\epsilon, \frac{1}{T} - \|\phi(s_{t+1}) - \phi(s_t)\|)$.

After optimizing $\phi$, the skill $z$ is aligned with the representation of the states. To utilize the magnitude of $z$, we assume that agent $\pi_{\theta_1}(\cdot \mid z)$ eventually arrives at the vector $z$ in the representation space. Therefore, the intrinsic reward is designed based on the relative distance as follows:

$$r_\phi(s_t, s_{t+1}, z) = \|z - \phi(s_t)\| - \|z - \phi(s_{t+1})\|. \tag{13}$$

Finally, $\pi_{\theta_1}(a \mid s, z)$ is optimized using the objective of SAC with the reward $r_\phi(s_t, s_{t+1}, z)$ at each learning stage.

### 3.3. Regret-aware Skill Generator for Skill Generation

According to Eq. 8, the objective of RSG is to find $z$ to maximize the regret. Using the policy gradient (Sutton et al., 1999; Silver et al., 2014) framework, we consider the distribution of $z$ as the action, and the regret serves as the value of the policy. $\pi_{\theta_2}^k$ is initialized at each stage before optimization. The objective of $\pi_{\theta_2}$ at stage $k$ is written as:

$$\max_{\theta_2} \ J(z) = \mathbb{E}_{z \sim \pi_{\theta_2}^k} \left[ Reg_k(z) \right]. \tag{14}$$

We assume that a skill distribution may not be fully mastered in a single stage. To stabilize the learning process, we maintain a population of skill generators, denoted as $P_z$, which can be formulated as follows:

$$P_z^k = \left\{ \pi_{\theta_2}^{k-i} \mid i = 0, \ldots, \min(l, k) \right\}, \tag{15}$$

where $l$ denotes the maximum size of $P_z$, and $\pi_{\theta_2}^{k-i}$ is the historical $\pi_{\theta_2}$ at stage $k - i$. The probability of $z$ given $P_z^k$ is given by: $p(z \mid P_z^k) = \mathbb{E}_{\pi_{\theta_2}^j \sim U(P_z^k)} \left[ \pi_{\theta_2}^j(z) \right]$, where $U(P_z^k)$ denotes that $\pi_{\theta_2}^j$ is uniformly sampled from $P_z^k$.

Although regret could reflect the converge strength, the regret estimate itself inevitably contains bias. Especially for

unseen $z$, the bias can mislead RSG, which would sample skills far from the frontier of the agent's capability. This issue is analogous to the challenge of maintaining stability of policy updates in RL. For instance, Proximal Policy Optimization (PPO) (Schulman et al., 2017) mitigates excessive policy deviation by employing a clipped advantage technique. Inspired by PPO, we propose a skill proximity regularizer to measure the distributional distance between the newly learned skill distribution and the previous ones.

To ensure diversity for skill learning, we first introduce a regularization to ensure that $\pi_{\theta_2}^k$ is different from $P_z$. This term measures the distributional distance between the newly learned distribution $\pi_{\theta_2}^k$ and the current $P_z$, to avoid adding redundant skills. We use KL divergence to estimate the diversity of the skill distributions as follows:

$$d_z = D_{\text{KL}} \left[ p(z \mid P_z^k) \| \pi_{\theta_2}^k(z) \right]. \tag{16}$$

Furthermore, we introduce another regularization to ensure that $\pi_{\theta_2}^k$ remains close to the frontier of the agent's capability. If $\pi_{\theta_2}^k$ is too far from the distribution of seen states, learning the new skill becomes too difficult. To mitigate this uncertainty, we maintain a buffer $\mathcal{B}_k$ that only stores the representation of states observed during each stage $k$. Since the distribution of $\mathcal{B}_k$ is generally not a standard distribution, we cannot directly compute the KL divergence as in $d_z(z)$. Instead, we focus on the nearest point in $\mathcal{B}_k$ relative to $\pi_{\theta_2}^k$. If the probability of the nearest point to $\pi_{\theta_2}^k$ is high, the distance between the point and $\pi_{\theta_2}^k$ is small, indicating that the uncertainty of the distribution is low and vice versa. Therefore, the regularization $d_\phi$ can be defined as follows:

$$d_\phi = \max_{\phi_{\text{seen}} \sim \mathcal{B}_k} \log \pi_{\theta_2}(\phi_{\text{seen}}), \tag{17}$$

where $\phi_{\text{seen}}$ represents the state representations from the buffer at stage $k$. We are going to increase $d_\phi$ to ensure that the new distribution is close to the distribution of seen states.

Finally, we define the positive constants $\alpha_1$ and $\alpha_2$ as weights for $d_z$ and $d_\phi$, respectively. Therefore, the objective function to optimize $\pi_{\theta_2}$ can be summarized as follows:

$$\max_{\theta_2} \ J_{\theta_2}(z) = \mathbb{E}_{z \sim \pi_{\theta_2}^k} \left[ Reg_k(z) \right] + \alpha_1 d_z + \alpha_2 d_\phi. \tag{18}$$

## 4. Experiments

### 4.1. Environments and Baselines

We compare our method with baselines on ant environment from dm_control (Tunyasuvunakool et al., 2020), Maze2d-large, Antmaze-medium, and Antmaze-large from D4RL (Fu et al., 2020). The Ant environment is used in LSD and METRA to demonstrate that the agent can learn

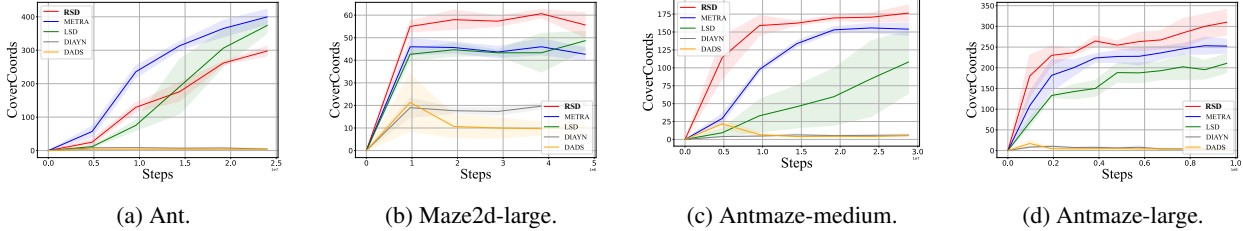

| (a) Ant. | (b) Maze2d-large. | (c) Antmaze-medium. | (d) Antmaze-large. |

*Figure 3.* Comparison between RSD and baselines, represented as red curves in different environments. The x-axis shows timesteps of interaction, while the y-axis represents Unique Coordinates, which measure state coverage achieved through sufficiently sampled skills.

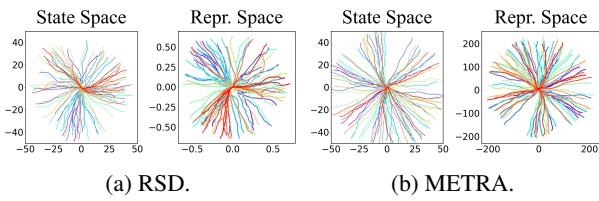

| (a) RSD. | (b) METRA. |

*Figure 4.* Visualization of Dynamical skills learned in Ant environment. Trajectories in State Space is shown on the left, and trajectories in Representation Space (Repr. Space) on the right.

temporal dynamic skills. The Ant environment contains an ant robot moving in an open map without obstacles, where exploration in any direction is nearly equivalent. We refer to this type of environment as a skill-symmetric environment. In contrast, exploration in environments with obstacles should vary in directions, which we refer to as skill-asymmetric environments. Maze2d-large has a lower state dimension compared to the other environments. The Antmaze-medium map is smaller and less complex than the Maze2d-large and Antmaze-large maps. We use these environments to demonstrate the performance of our method in a high-state dimension and more complex map.

We primarily compare our method with the baselines ME-TRA, LSD, DIAYN, and DADS. DIAYN is one of the first methods to highlight the potential of diverse skill discovery to solve general tasks based on mutual information reformulated with entropy. DADS prioritizes discovering predictable behaviors while simultaneously learning the dynamics of the environment. LSD further introduces a novel Lipschitz representation space to discover temporal dynamic skills. METRA builds on LSD and mathematically establishes the relationship between its objective and PCA-based clustering, demonstrating its scalability. We compare our RSD with the baselines to demonstrate that ours retains the advantages of METRA while improving learning efficiency and state coverage in skill-asymmetric environments. Other USD methods, such as (Gregor et al., 2016; Laskin et al., 2022; Bae et al., 2024), also use maze environments to test their diversity. However, they either fail to learn dynamic skills in high-dimensional environments or rely on additional exploration intrinsic rewards. In contrast, our method

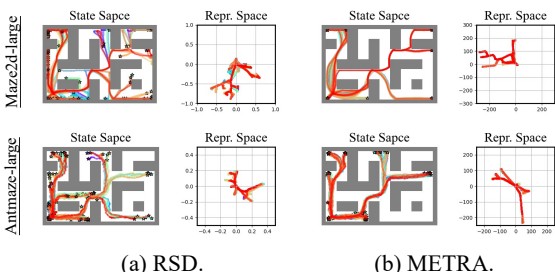

| (a) RSD. | (b) METRA. |

*Figure 5.* The visualization of state and representation coverage in Maze2d-large and Antmaze-large. The results of RSD are shown on the left, and the results of METRA are on the right.

can improve efficiency simply by adaptively sampling skills, a strategy that has been overlooked in previous work.

When evaluating, we thoroughly sample the $z$ in the latent skill space $Z$ to ensure that they cover the full range of skills. We then assess state coverage by recording the unique coordinates visited on the map, referred to as CoverCoords, to evaluate the diversity of skills. A higher value of CoverCoords indicates higher state coverage across all skills, which means that the skill set is more diverse. In particular, our RSG learns a Gaussian distribution at each stage, making $P_z$ a Gaussian Mixture Model (GMM). Additionally, we design a greedy adaptive sampling technique during the agent's learning process. A more detailed experimental setup and techniques of our method can be found in the appendix C.

### 4.2. Experiment Analysis

Figure 3 shows the curves of CoverCoords over timesteps. Our method demonstrates competitive results in skill-asymmetric environments. According to Figure 3a, we demonstrate that RSD has the ability to learn dynamic temporal skills in different directions, compared to DIAYN and DADS. Also, the curves show that the state coverage of RSD is lower than LSD and METRA. This outcome in skill-asymmetric environments can be attributed to two key factors: First, in baseline methods, the skill $z$ is represented as a unit vector, lacking magnitude information. In contrast, our approach samples $z$ from the entire space $Z$, requiring the learning of a broader range of skills compared to previ-

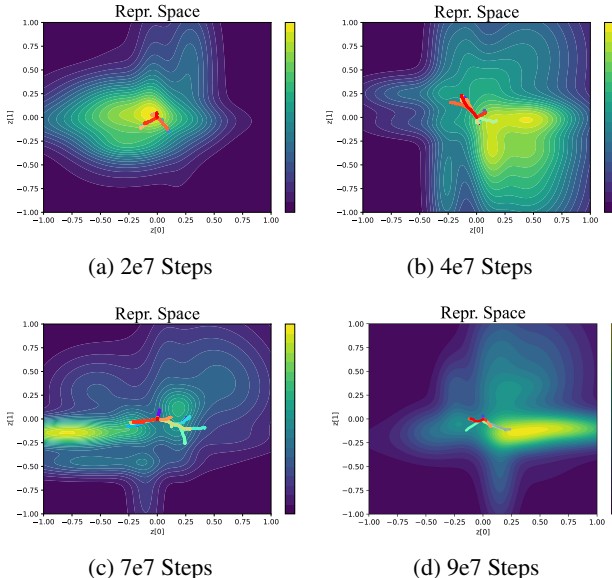

(a) 2e7 Steps       (b) 4e7 Steps

(c) 7e7 Steps       (d) 9e7 Steps

*Figure 6.* Filled contour plot of the population of skill generators in representation space across different training stages. Color-coded lines depict distinct skill trajectories sampled according to $P_z$.

ous methods within the same number of timesteps. Second, in skill-symmetric environments like Ant, every direction is equally potential, making uniform sampling the most effective approach. However, our method can still learn sufficiently diverse directions, producing results similar to those of METRA, as shown in both spaces in Figure 4.

As shown in Figures 3b, 3c, and 3d, our approach achieves greater efficiency and greater state coverage in all three environments. In these skill-asymmetric environments, DIAYN and DADS struggle to explore beyond the first corner of the map, resulting in low coverage. The reason is that, although these previous methods effectively differentiate and cluster skills in the latent space, they struggle with long-range exploration and fail to learn temporally dynamic skills. Both METRA and LSD perform reasonably well in Maze2d-large environment, where the state space has relatively low dimension. However, as the state dimension increases in the following environments, the efficiency of the LSD methods decreases. In comparison, our method in these environments not only has greater learning efficiency, but also achieves higher state coverage without incorporating additional intrinsic exploration rewards, such as RND (Bae et al., 2024).

We observe that the baseline method tends to explore primary directions, consistent with the PCA-like theoretical explanations provided in their paper (Park et al., 2024c). However, in skill-asymmetry environments, this feature leads to homogeneous skills, resulting in a less diverse skill set. In contrast, our method employs non-unit vector $z$ and non-uniform sampling strategies. The non-unit vector helps to distinguish the points behind barriers, and the non-uniform sampling strategies allow for a more focused exploration

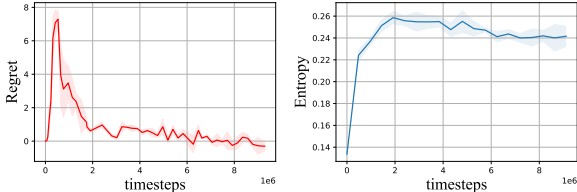

*Figure 7.* The curves of Regret and Entropy over timesteps in Maze2d-large. Regret is calculated using Eq. 7 after updating the $\pi_{\theta_1}$. And the Entropy denotes the entropy of the $P_z$ after updating the $\pi_{\theta_2}$, estimated by a Monte Carlo method.

of similar but different skills. Figure 5 visualizes why our algorithm enhances state coverage in these environments. Our method is capable of discovering skills to more corners in maze maps, and its learned representations exhibit more discriminative details compared to those of METRA.

To further demonstrate this, we present contour maps of the skill distributions $P_z$ at different learning stages. In order to further highlight the learning process related to the skill population $P_z$, we also plot the trajectories in the representation space on the map, as shown in Figure 6. A direct observation is that $P_z$ differs between different learning stages. This shows that $P_z$ is capable of sampling in various directions following the objective of Eq. 18. As shown in Figure 6b, a single $P_z$ may exhibit multiple peaks, indicating the presence of valuable skills to learn at the same time. These adjacent peaks can serve as a form of contrastive learning for similar behaviors, which explains why RSD learns more detailed skills. Further results in Figure 6c confirms that the agent actually learns the corresponding skills with a high probability in Figure 6b after training subsequently. Specifically, this region corresponds to the branching area in the upper-right corner of the lower-right corner on the Antmaze-large map.

Figure 7 depicts the trends of the Regret calculated based on Eq. 7 and the Entropy of $P_z$ in Maze2d-large environments. The sharp rise of Regret demonstrates that $P_z$ first identifies the skills that maximize regret, presumed to be the agent's capability boundary. Meanwhile, the entropy increases, indicating that the skill population becomes more diverse. Subsequently, the Regret begins to decrease, while the entropy remains stable. This shows that the agent has mastered a wider set of skills. Finally, the regret diminishes to zero, suggesting that further learning yields no additional benefits. This trend also serves as evidence that our algorithm is capable of achieving convergence.

To better understand the contribution of each module, we conduct ablation studies, with detailed results presented in Appendix E. Additionally, we evaluate our method in the kitchen environment, utilizing a discrete skill settings alongside our proposed skill generation policy. Further details can be found in Appendix F.

## 4.3. Zero-shot Performance

We use navigation tasks as downstream tasks to evaluate the zero-shot capability of our method. During the evaluation process, the agent is provided with a specific goal state $s_g$, without any further training process or any external signals. We directly measure the agent's final distance (FD) to the goal point and the success rate (AR) under the corresponding $z_g$. For baselines, $z_g$ is defined as the unit vector of the difference between the representation space vectors of $s_g$ and $s_0$. For our algorithm, $z_g$ corresponds to the vector representation of $s_g$. We selected Maze2d-large and Antmaze-large environments for comparison, as they share similarly shaped maps but differ in state dimensionality. We uniformly sampled goal points from each map to serve as $s_g$, with detailed visualizations provided in Appendix D.

As shown in Table 1, RSD achieves the highest success rates (AR) and the smallest final distances (FD) in both environments. These results demonstrate that our algorithm has zero-shot capability after unsupervised exploration and skill learning, similar to previous work. Furthermore, the improvement in AR is more pronounced in the higher-dimensional Antmaze-large environment, whereas the reduction in FD is less significant compared to the lower-dimensional Maze2d-large environment. This discrepancy arises because the baseline can already explore most relatively reachable points in Maze2d-large. The improvement in FD is less significant in high-dimensional environments because $s_g$ is more likely to fall out of distribution, leading it to misinterpret the goal and select suboptimal or incorrect directions at the beginning. This mismatch explains why the reduction in FD is less pronounced in high-dimensional environments compared to low-dimensional counterparts.

## 5. Related Work

### 5.1. Unsupervised skill discovery

Unsupervised skill discovery (USD) is an important subfield of RL exploration based on information theory (Hao et al., 2023; Ladosz et al., 2022). USD addresses unsupervised exploration and learning useful behaviors (Gregor et al., 2016), enabling rapid adaptation to downstream tasks. As one of pioneering works, DIAYN (Eysenbach et al., 2019) demonstrates the feasibility that the discovering diverse skill improves downstream task performance. They employ mutual information conditioned on skill vectors and optimize a variational lower bound as their objective. Later, CIC (Laskin et al., 2022) maximize the entropy by employing contrastive learning between states and skills, enhancing the diversity of learned skills. While Yang et al. (Yang et al., 2023) also successfully leverage the advantage of contrastive learning, their method fails in high-dimensional settings and suffers from unstable training due to conflicting objectives. Mazza-

Table 1. Performance evaluation under the goal-condition tasks. The bolded data represents the best performance for the corresponding metric. Metra-f (Metra-fixed) represents the setting where $z$ remains fixed after initialization. Metra-d (Metra-dynamic) represents the setting where $z$ is recalculated at each timestep.

| | Maze2d-large | | Antmaze-large | |
|---|---|---|---|---|
| | FD↓ | AR↑ | FD↓ | AR↑ |
| DIAYN (Eysenbach et al., 2019) | 4.249 | 0.130 | 23.7 | 0.043 |
| DADS (Sharma et al., 2020) | 4.972 | 0.073 | 23.8 | 0.043 |
| LSD (Park et al., 2022) | 2.843 | 0.304 | 8.12 | 0.362 |
| METRA-f (Park et al., 2024c) | 3.335 | 0.333 | 9.90 | 0.450 |
| METRA-d (Park et al., 2024c) | 0.658 | 0.782 | 6.76 | 0.435 |
| RSD(Ours) | **0.535** | **0.811** | **6.42** | **0.507** |

glia et al.(Mazzaglia et al., 2022) achieves skill discovery through model-based RL, which is more computationally expensive and requires additional exploration strategies.

However, previous works like (Eysenbach et al., 2019; Sharma et al., 2020; Gregor et al., 2016) does not account for temporal dynamics in skill learning in high-dimension environments, as pointed by Park, S et al. (Park et al., 2022). To address this limitation, they propose LSD to address this limitation by introducing temporal representation learning in Lipschitz space. METRA (Park et al., 2024c) further focuses on mutual information between the last states and skill variables, and utilizes Wasserstein dependency to decompose the objective. They demonstrate that METRA is an approach analogous to PCA clustering, and is reliable in high-dimensional spaces. However, these methods typically sample skills uniformly, overlooking the relation between skills and the learning policy in different environments.

### 5.2. Automatic curriculum learning

Automatic curriculum learning (ACL) is a mechanism that leverages the learning status of the agent's capabilities to adaptively change the distribution of training data (Portelas et al., 2020). Generally, ACL have multiple explicit tasks or goals to learn, which is widely used in robotic manipulation (Fournier et al., 2018; Fang et al., 2019), or multi-goal navigation (Sukhbaatar et al., 2018; Florensa et al., 2018; Colas et al., 2020). Another purpose of ACL is to facilitate the open-ended exploration (Pong et al., 2020; Jabri et al., 2019; Colas et al., 2020), where the distribution of achievable goals is initially unknown. Among these, PLR (Jiang et al., 2021b) is most related to our topic, identifying the limitation of random sampling of training levels independently of the agent policy. They introduces scoring functions based on GAE (Schaul et al., 2016) to indicate the learning status across varying environments. Subsequent work (Jiang et al., 2021a) has expanded on this foundation. They propose teacher-student dual-policy architectures to handle with more difficult maps. (Parker-Holder et al., 2022) also uti-

lizes a min-max regret policy to design environment for improving the robustness of the RL agent. In other application domains, adversarial learning frameworks (Li et al., 2023; 2024) have also demonstrated effectiveness in enhancing model robustness against out-of-distribution data scenarios. However, these ACL approaches rely on explicit rewards, making them unsuitable for unsupervised discovery scenarios which lack stable reward signals. Our method distills insights from them and leverage them for efficient unsupervised skill discovery.

## 6. Conclusion

In this paper, we argue that the skill discovery process is influenced by policy converged strength. Based on this insight, we proposed a min-max adversarial algorithm (named RSD) which comprises an agent policy and a RSG to enhance learning efficiency. By further incorporating skill proximity regularizer and an updatable skill population, our method effectively balances performance and diversity within a bounded temporal representation space. Experimental evaluations demonstrated the effectiveness of our approach, highlighting improvements in sample efficiency, skill diversity, and zero-shot abilities, particularly in skill-asymmetrical scenarios. These findings underscore the potential of our method to address challenges in efficient exploration without relying on additional intrinsic exploration rewards. For future work, we aim to integrate our framework with hierarchical RL architectures, enabling efficient applications in the scenarios that involve asymmetric skills.

## Acknowledgements

This work was supported in part by the National Key R&D Program of China (Grant No. 2023YFF0725001), in part by Shanghai Artificial Intelligence Laboratory and National Key R&D Program of China (2022ZD0119302), in part by the National Natural Science Foundation of China (Grant No. 92370204, 62306255), in part by the Guangdong Basic and Applied Basic Research Foundation (Grant No. 2023B1515120057, 2024A1515011839), in part by Guangzhou-HKUST(GZ) Joint Funding Program (Grant No. 2023A03J0008), the Fundamental Research Project of Guangzhou (No. 2024A04J4233), and Education Bureau of Guangzhou Municipality.

## Impact Statement

This paper advances unsupervised skill discovery in reinforcement learning by improving both efficiency and diversity in skill acquisition. Potential applications include real-world manipulation tasks involving sparse rewards and open-ended challenges, offering benefits such as reduced computational costs and improved adaptability. We do not foresee any ethical concerns associated with this work.

## Software

Our code is open-source at `https://github.com/ZhHe11/RSD`.

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

## A. Additional Related Work

### A.1. Other technique lines for unsupervised skill discovery

Recent works (Bae et al., 2024; Bai et al., 2024; Kim et al., 2024) attempt to improve the skill diversity by introducing additional intrinsic rewards for exploration such as prediction error models (Burda et al., 2019; Pathak et al., 2017) or K-nearest neighbors, facing limitations in high-dimensional spaces. In contrast, we argue that efficiency can be improved simply by adaptively sampling strategies according to the policy regret. Recent works also explored learning skills using successor features in dynamic environments (Lee & Seo, 2023) or proposed triple-hierarchical decision frameworks for downstream tasks (Wang et al., 2024a). However, they did not fully explore the skills within an environment and overlooked the importance of skill policies for efficiency improvement.

### A.2. Goal-conditional RL with temporal representation

Goal-conditional reinforcement learning (GCRL) has flourished in recent years due to its potential for multitask generalization with sparse rewards (Andrychowicz et al., 2017; Kaelbling, 1993; Ding et al., 2019; Ghosh et al., 2019; Nasiriany et al., 2019; Sun et al., 2019; Eysenbach et al., 2020; Liu et al., 2024). This area is also important because it can enhance the interpretability of learned models (Ji et al., 2025). An approach to providing continuous and dense intrinsic rewards is to leverage the temporal corelation between goals and states. For instance, QRL (Wang et al., 2023) introduces the use of quasimetric to reconstruct the distance in a temporal representation space. HIQL (Park et al., 2024a) acquires feasible subgoals by utilizing the vectors between goals and states in the temporal representation space, which benefits long-horizon tasks. Later, HILPS (Park et al., 2024b) proposed an unsupervised offline training method, based on learning the temporal representation in a Hilbert space first. Although these approaches perform well in offline GCRL settings, their effectiveness in online unsupervised RL scenarios is limited. Instead, we build upon problem formulation in GCRL and incorporate the concept of temporal representation learning for the discovery of unsupervised skills, as suggested in previous work (Park et al., 2024c).

ji2025comprehensive

## B. Environment Description

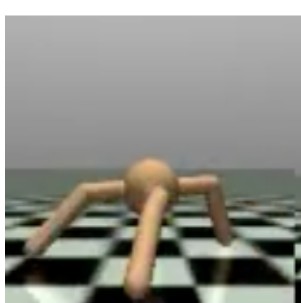 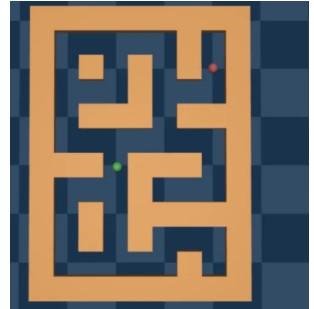 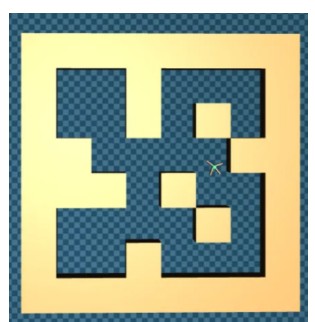 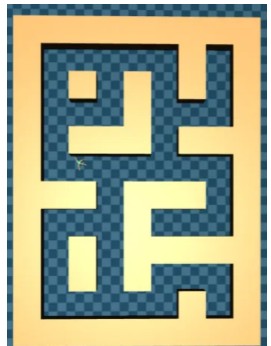

*Figure 8.* Examples of the environments

Our research encompasses four environments: Ant, maze2d-umaze-dense-v1, antmaze-medium-diverse-v0, and antmaze-large-diverse-v0, as illustrated in Figure 13. Specifically, the state dimension of Ant is 29, Maze2d is 4, and Antmaze is also 29. In particular, the Antmaze-Large map resembles that of Maze2d-Large in map shape but differs in scale, which explains the varying FD values in Table 1. Many studies (Bagaria et al., 2021; Campos et al., 2020; Kim et al., 2024; Bae et al., 2024) on USD utilize the maze environment, to demonstrate the diversity of skills and their effectiveness in solving downstream tasks. Our experiments are similarly designed to validate the efficacy of our method based on these two criteria. While these methods have achieved diversity in the maze environment, as previously noted, they exhibit certain limitations. Some methods succeed only in low-dimensional spaces, such as Maze2d, but fail in environments like Ant or Antmaze due to the lack of temporal representation learning. Others rely on additional unsupervised exploration techniques, such as RND, which introduce instability and expose weaknesses associated with model prediction errors, such as the inability to handle the white noise scenarios, which contains diverse but meaningless states.

# C. Experimental Setup

## C.1. Parameter Setting

| HYPERPARAMETER | VALUE |
|---|---|
| MAX PATH LENGTH | 300 |
| TRAJECTORY BATCH SIZE | 16 |
| SAC MAX BUFFER SIZE | 3000000 |
| OPTION DIM | 2 |
| LEARNING RATE (common) | 0.0001 |
| LEARNING RATE ($\phi$) | 0.001 |
| MODEL LAYER | 2 |
| MODEL DIM | 1024 |
| DISCOUNT FACTOR | 0.99 |
| BATCH SIZE | 1024 |
| DUAL SLACK | 0.001 |
| $\alpha_1$ ($\pi_{\theta_2}$) | 5 |
| $\alpha_2$ ($\pi_{\theta_2}$) | 1 |
| MAX SIZE ($P_z$) $l$ | 15 |
| STEPS IN EACH STAGE | 50 |

This shows the parameters in Maze2d-large environment. To ensure fairness, the parameters shared between the baseline and our method were kept consistent. All experiments were conducted on a single NVIDIA A100 GPU with PyTorch 2.0.1.

## C.2. Explanation for Learning Process

Our method offers aligns closely with the foundational theories of the Learning Process. These theories suggest that the Learning Process (Kaplan & Oudeyer, 2007) can serve as an essential guide for exploration, effectively replacing random exploration strategies. We base our approach on the following two assumptions: 1) The difficulty of skills varies depending on the environment. 2) Skills are interrelated, with basic skills learned before advanced ones. By analyzing successive Learning Progress (Gottlieb & Oudeyer, 2018; Kim et al., 2020) through the lens of converged proficiency, our method addresses these two assumptions: 1) sample fewer easy skills that have already shown no further progress; and 2) sample fewer advanced skills, which also show no progress until the agent masters the basic ones".

## C.3. Experiment Techniques

For the skill population, we employ three techniques in practical implementation:

### C.3.1. CENTERING REPRESENTATIONS

To ensure that all directions in the representation space are fully utilized, we introduce a constraint to center the representation of the initial state $s_0$. This is achieved by enforcing $\phi_0 = \vec{0}$. Based on Eq. 3.2, we add an additional constraint as follows:

$$C_0(s_0) = -\|\phi(s_0)\|.$$

### C.3.2. HEURISTIC POPULATION UPDATE

We adopt a heuristic method to update the skill population efficiently. Specifically, if the current population size $|P_z|$ reaches its maximum limit $l$, we remove the component with the lowest regret value. This approach allows for more efficient utilization of the population and reduces the probability of learning redundant skills.

### C.3.3. GREEDY ADJUSTMENT OF POPULATION DISTRIBUTION

During each learning stage at Part 1, we adjust the distribution of the skill population $P_z$ using a greedy method. In our implementation, $P_z$ is modeled as a Gaussian Mixture Model (GMM), where each component corresponds to a Gaussian distribution $\pi_{\theta_2}$. Adjusting the sampling probability of each component is feasible. We leverage the $\phi_{\text{seen}}$ values, which is formulated during the computation of the distance regularizer $d_\phi$ 17. We consider the final state $s_f$ in new collected trajectories. Then, the representation of $s_f$ is denoted as $\phi_{s_f}$. If $\phi(s_f)$, compared to $\phi_{\text{seen}}$, is closer to a specific component in $P_z$, it indicates that the corresponding distribution is being effectively learned. Consequently, we increase the sampling probability of that component. The GMM's component probabilities are recalibrated using a softmax function:

$$w_i = \max_{\phi_{s_f} \sim \mathcal{B}_{\text{new\_traj}}} \left(\log(\pi_{\theta_2}(\phi_{s_f}))\right) - \max_{\phi_{s_f} \sim \mathcal{B}_{\text{old\_traj}}} \left(\log(\pi_{\theta_2}(\phi_{s_f}))\right),$$

$$P(i) = \frac{\exp(w_i)}{\sum_j \exp(w_j)},$$

where $w_i$ is the weight associated with the $i$-th component, $\mathcal{B}_{\text{new\_traj}}$ and $\mathcal{B}_{\text{old\_traj}}$ represent the trajectories collected at two successive steps. To ensure fairness and diversity, we enforce a minimum probability threshold for each component, guaranteeing that every component has a chance of being sampled.

## D. Visualization of Zero-shot Evaluation

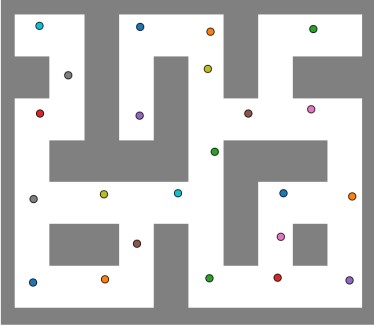

*Figure 9.* Goals for zero-shot evaluation.

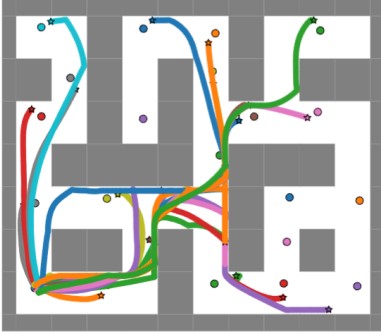

*Figure 10.* Visualization of our zero-shot performance.

Figure 9 shows the goals we used for zero-shot evaluation in Maze2d-large environment, which are evenly distributed across the reachable areas of the map. Figure 10 shows the trajectories of our method's performance on the map. We consider the agent to have reached the goal if the L2 distance is less than 1.

# E. Hyperparameter Tuning

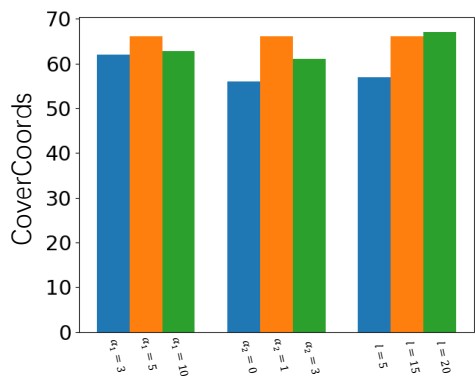

*Figure 11.* Hyperparameter Tuning of $\alpha_1$, $\alpha_2$ and maximum length $l$.

We aim to analyze the impact of the hyperparameters $\alpha_1$, $\alpha_2$, and the maximum length $l$ on the performance of our model. In Figure 11, each cluster of results corresponds to the adjustment of a single parameter, while keeping the other parameters fixed.

The parameter $\alpha_1$, which controls the diversity of $P_z$. When $\alpha_1$ is too large, the policy $\pi_{\theta_2}$ tends to ignore the regret values and prioritize diversity, leading to behavior similar to uniform sampling. In this case, the value of regret are not utilized effectively. Conversely, when $\alpha_1$ is too small, the policy may repeatedly sample skills from similar regions, resulting in redundant skill exploration.

The parameter $\alpha_2$, which prevents $P_z$ from venturing too far from the seen regions. When $\alpha_2$ is set to 0, we observe that $\pi_{\theta_2}$ frequently selects unseen $z$ values for learning. This is due to the regret estimation function inaccurately assigning high regret values to unseen $z$, which the agent is not yet equipped to handle effectively. On the other hand, if $\alpha_2$ is too large, the policy tends to sample already explored points excessively, reducing the probability of exploration, and thereby hindering the model's efficiency.

Lastly, the maximum length $l$ of $P_z$. Increasing $l$ provides more steps for skill learning and helps prevent forgetting, thereby improving performance. However, further increasing $l$ yields diminishing returns. Based on our observations, we choose $l = 15$ as a balanced value that ensures effective learning.

# F. Kitchen Environment Evaluation

### F.1. Motivation for Kitchen Selection

To further validate our method's effectiveness, we conducted additional experiments in the Kitchen environment, known to present a significant challenge in skill discovery. Figure 5 illustrates the performance comparison between METRA and our proposed method in this challenging scenario. Notably, METRA struggles in this environment due to skill asymmetry, an issue specifically targeted by our approach. To test our mehtod's ability on pixel-based environment, we extended our experimental assessment to the Kitchen environment, characterized by pixel-based observations and complex robotic manipulation tasks, as previously explored by METRA.

### F.2. Experimental Setup

Consistent with METRA's established methodology, performance evaluation was based on the completion of a set of 7 predefined tasks. We report the mean and standard deviation (mean ± std) across three random seeds 0, 2, 4. Experiments were conducted at varying training stages (100k, 200k, 300k, and 400k steps) and with two different skill dimensions (24 and 32).

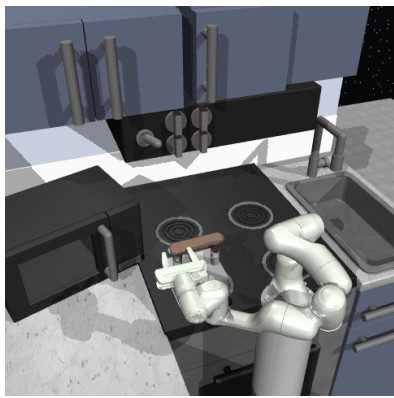

*Figure 12.* Example of Kitchen Environment.

## F.3. Results at different Skill Dimensions

**Results at Skill Dimension = 24**: Our method significantly outperformed METRA, particularly at later training stages. At 400k steps, our method achieved 5.08±0.45 completed tasks compared to METRA's 3.94±0.36, showing a clear improvement of +1.14 tasks.

| Model | 100k | 200k | 300k | 400k |
|-------|------|------|------|------|
| METRA | 2.72±0.19 | 3.20±0.27 | 3.93±0.67 | 3.94±0.36 |
| Ours | 2.41±0.02 | 3.61±0.02 | 4.29±0.11 | **5.08±0.45** |

*Table 2.* Kitchen Environment Performance (Skill Dimension=24)

**Results at Skill Dimension = 32**: At a higher skill dimension, our approach continued to surpass METRA. At 400k steps, we obtained 4.55±0.35 completed tasks against METRA's 3.99±0.94, marking an improvement of +0.56 tasks.

| Model | 100k | 200k | 300k | 400k |
|-------|------|------|------|------|
| METRA | 3.21±0.71 | 3.66±0.69 | 3.83±0.90 | 3.99±0.94 |
| Ours | 3.51±0.23 | 3.91±0.43 | 4.34±0.39 | **4.55±0.35** |

*Table 3.* Kitchen Environment Performance (Skill Dimension=32)

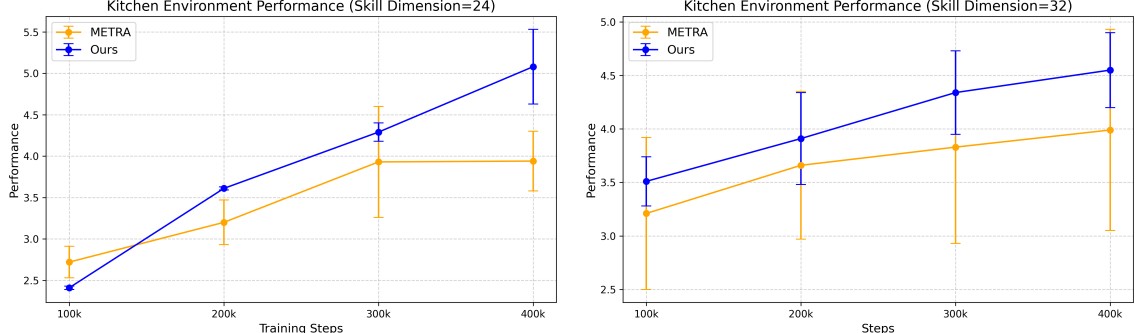

*Figure 13.* Examples of the environments

Overall, our method consistently demonstrated superior sample efficiency and final performance compared to METRA. Particularly at a skill dimension of 32, the performance gains were increasingly pronounced as training progressed (e.g., at 300k steps was +0.51 compared to +0.36 at dimension 24). These improvements are clearly visible in performance curve visualizations, underscoring our method's robustness and scalability.

