# OpenReview forum: "Efficient Skill Discovery via Regret-Aware Optimization"
_ICML.cc/2025/Conference — ICML 2025 poster_

### Official Review · Reviewer_aHNd · 2025-02-18

**Overall Recommendation:** 3

**Summary:**

This paper proposes regret-aware skill discovery (RSD) for unsupervised skill discovery. RSD is built upon METRA, a previous temporal distance-based skill discovery method. The key idea behind RSD is to use a separate, learned skill sampler policy to sample $z$'s for better exploration (unlike the uniform distribution in METRA). Specifically, they train two policies in an adversarial manner: an action policy $\pi_{\theta_1}$ and a skill sampler policy $\pi_{\theta_2}$. $\pi_{\theta_1}$ maximizes an objective similar to METRA (but with slight modifications to make it compatible with bounded $\phi$). $\pi_{\theta_2}$ maximizes the regret (defined as $V_k - V_{k-1}$) of the action policy to guide the agent into a region where there's room for improvement in skill learning. The authors experimentally show that RSD leads to better state coverage and downstream performance in Ant, Maze2d, and AntMaze domains.

**Claims And Evidence:**

Their claims made in the abstract and introduction are generally well-supported by empirical evidence.

**Essential References Not Discussed:**

One particularly related missing work is "TLDR: Unsupervised Goal-Conditioned RL via Temporal Distance-Aware Representations" by Bae et al. (2024). Their method is also based on METRA and is evaluated mainly on AntMaze. While I don't believe the comparison with this method is necessary in assessing this paper, it'd have been better if the authors had (at least) discussed this work in the related work section.

**Experimental Designs Or Analyses:**

They use standard tasks in unsupervised RL (Ant, PointMaze, and AntMaze), which look reasonable to me.

**Methods And Evaluation Criteria:**

Their approach is sensible and the evaluation criteria look reasonable to me.

**Other Comments Or Suggestions:**

I spotted quite a lot of grammatical errors and inconsistencies throughout the paper. I'd recommend running a spell/grammar checker. To list only a few:
* L145: "for agent policy" -> "for the agent policy"
* L148: "These improvements ensures" -> "These improvements ensure"
* L245: "denotes maximum size of" -> "denotes the maximum size of"
* L7 of Algorithm 1: Why is $\langle, \rangle$ used instead of $(, )$ to denote tuples as in L72?
* Equation (2) uses $L$ but Equation (3) uses $\mathcal{L}$. Moreover, this notation is not defined (and it's Lipschitz w.r.t. which metric?).

**Other Strengths And Weaknesses:**

### Strengths
* The concept of using a learned skill sampling distribution seems (relatively) novel to me (at least it has not been well-studied in the literature). The use of regrets seems quite sensible to me as well.
* Figure 5 is particularly convincing to me. The authors convincingly demonstrate that RSD leads to better diversity and state coverage than METRA by having a non-uniform skill sampler.
* The paper provides diverse analyses studying different aspects of RSD training.


### Weaknesses
* In my opinion, the main weakness of this paper is writing.
  * Section 3 does not flow very well. It directly starts with the detailed description of $\pi_{\theta_1}$ and $\pi_{\theta_2}$ and presents the full algorithm box without motivating or defining the notations. In particular, the "types" of the policies are never defined -- I later realized that $\pi_{\theta_1}$ outputs actions and $\pi_{\theta_2}$ outputs $z$s, which confused me a bit.
  * Equation (8) is also quite confusing. It is unclear how $\theta_2$ affects the objective at this point -- it turns out only in Section 3.3 that $P_z$ depends on $\pi_{\theta_2}$. Also, why does $\pi_{\theta_1}$ minimize this objective? Isn't it also supposed to maximize $V_{\pi_{\theta_1}}^k$ (treating $V_{\pi_{\theta_1}}^{k-1}$ as a constant)?
  * I believe Section 3.2 and Section 3.3 should come before Section 3.1 (or at least the paper should be heavily restructured in general) as $Q$ and $V$ depend on the reward function defined in Section 3.2.
  * Equation (11) needs a better explanation. While I understood this, it might be unclear why "a well-learned skill trajectory in the bounded space must hold" the equation below to the general reader.
  * L193: "Due to the bounded nature of $\phi( \cdot )$, it naturally satisfies the Lipschitz constraint.": Boundedness and Lipschitzness are separate concepts. For example, the Heaviside step function is bounded but not Lipschitz (and not even continuous).
* Regarding experiments, the domains are limited to PointMass and Ant. In its current form, it is unclear how RSD scales to pixel-based observations or other types of environments (e.g., whole-body humanoid control, robotic manipulation, etc.).

**Questions For Authors:**

* Is there a mechanism to prevent a collapse of $\pi_{\theta_2}$? For example, it can collapse to a single point (e.g., $(1, 1, \cdots, 1)$) to maximize $d_z$, and keeping this collapsed skill sampler policy for a while would maximize $d_\theta$ as well. If this specific skill leads to high regrets, then (hypothetically) the agent might end up learning only a single skill. Is there a reason why this doesn't happen in general?
* Unlike METRA, $z_\mathrm{updated}$ in Equation (10) depends on the current latent state $\phi(s_t)$. How does this affect skill learning? Can the authors elaborate on the pros and cons of this choice?

**Relation To Broader Scientific Literature:**

This paper tackles unsupervised skill discovery, and the main takeaway (to me) is that a non-uniform skill sampling distribution can lead to better exploration. This is often overlooked in the previous literature, as prior works mostly employ a symmetric, uniform distribution (Gaussian, uniform over a box, etc.). I believe this is a nice insight to the community.

**Theoretical Claims:**

N/A

---

> ### Author Rebuttal · Authors · 2025-03-30
>
> ## **Q1: Collapsing Concern**
> Thank you for raising this insightful question. Our method is specifically designed to prevent skill collapse. As shown in Eq. (15), the skills are maintained within a population $P_z$. We address the diversity concern (i.e., avoiding convergence to a single point) from three perspectives:
> 1. **KL divergence constraint (Eq. 16):**
>    We enforce a divergence between the newly generated skill distribution $\pi_{\theta_2}$ and existing skills in $P_z$ using KL divergence. This ensures new skills remain distinct.
> 2. **Minimum variance and overlap check:**
>    The sampler $\pi_{\theta_2}$ (Gaussian) is constrained to maintain a minimum std (1e-1). We also check for distributional overlap before adding new skills, preserving diversity.
> 3. **Empirical validation (Figure 7):**
>    Figure 7 shows a consistent decrease in average regret, mainly driven by SAC. As SAC improves, $\pi_{\theta_2}$ keeps adapting, avoiding collapse to a single mode.
>
> Please let us know if further clarification is needed!
> ## **Q2: State-dependent Skill $z_{\text{updated}}$**
> Thank you for your careful observation.
> - When $z$ is fixed across time, Eq. (4) tends to force straight-line trajectories. In skill-asymmetric environments, skills often share key states, making this problematic (Figure 5).
> - Allowing $z_{\text{updated}}$ to vary with $\phi$ and time $t$ enables distinct **trajectory curves**, even with shared segments, better capturing behavioral nuances.
>
> **Advantages:**
> Improves **skill diversity** (Figures 5 & 6), better covers complex regions (e.g., maze corners), and outperforms METRA in skill expressiveness.
>
> **Disadvantages:**
> Increases **learning complexity**, as seen in Figure 3(a), where RSD is slightly less efficient in simpler environments.
> ## **Discussion on Related Work**
> We appreciate the reviewer highlighting TLDR[Bae et al., 2024], which is cited in Lines 303, 329, and Appendix A.1 (L608). Our method focuses on skill sampling scheme, whereas TLDR relies on RND and KNN-based exploration strategies.
> We will clarify this in the revised related work section.
>
> ## **Weakness Response**
> ### 1. Clarity of Policies $\pi_{\theta_1}$ and $\pi_{\theta_2}$
> We revised the start of Section 3 using a bullet-point format to clearly distinguish. We believe this change helps readers immediately grasp the roles of each policy.
> ### 2. Organization of Section 3
> Our intention was to follow a “global-to-local” structure for clarity.
> -  Base reward is defined earlier in **Preliminaries (Eq. 4)**.
> - We were concerned that Keeping Section 3.1 ahead preserves motivation for why $\pi_{\theta_1}$ minimizes and $\pi_{\theta_2}$ maximizes regret.
> ### 3. Why $\pi_{\theta_1}$ Minimizes Regret (Eq. 8)
> Great point — we clarify:
> - Regret is estimated via Eq. 7.
> - If $\pi_{\theta_1}$ has converged for skill $z$, regret (value improvement) is near zero.
> - Thus, minimizing regret implies convergence under that skill.
>
> Note: $\pi_{\theta_1}$ is still trained with standard SAC loss (Eq. 13, Alg. 1 line 10).
> ### 4. Intuition Behind Eq. (11)
> For example, if $\||\phi(s_T)\|| \leq 1$ and we set $\phi(s_0) = 0$, then by telescoping:
> $ \|| \sum_t^{T-1} \phi(s_{t+1}) - \phi(s_t) \|| \leq 1$.
> To simplify, we enforce Eq. (11), and one solution is:  $\||  \phi(s_{t+1}) - \phi(s_t) \|| \leq 1/T$.
> ### 5. Additional Results
> We added experiments in the **Kitchen environment** (pixel observations, robotic control), following **METRA**’s setup and reporting success out of 7 tasks.
>
> As shown in the tables below, our method shows both **higher sample efficiency** and **stronger final performance**.
> Interestingly, at **dim = 32**, our method exhibits **faster efficiency gains** (e.g., Δ@300k = **+0.51**) compared to **dim = 24** (Δ@300k = **+0.36**), suggesting that our efficient approach benefits more from a larger latent space. This trend is even clearer when visualized in the learning curves.
> #### Performance @ Skill Dim = 24
> | **Model** | **Step = 100k** | **Step = 200k** | **Step = 300k** | **Step = 400k** | **Δ@400k** |
> |:-:|:-:|:-:|:-:|:-:|:-:|
> | *METRA*   | **2.72 ± 0.19**     | 3.20 ± 0.27     | 3.93 ± 0.67     | 3.94 ± 0.36  |  |
> | *Ours*    | 2.41 ± 0.02 | **3.61 ± 0.02** | **4.29 ± 0.11** | **5.08 ± 0.45** | **+1.14**  |
> #### Performance @ Skill Dim = 32
> | **Model** | **Step = 100k** | **Step = 200k** | **Step = 300k** | **Step = 400k** | **Δ@400k** |
> |:-:|:-:|:-:|:-:|:-:|:-:|
> | *METRA*| 3.21 ± 0.71| 3.66 ± 0.69| 3.83 ± 0.90| 3.99 ± 0.94     |    |
> | *Ours*| **3.51 ± 0.23** | **3.91 ± 0.43** | **4.34 ± 0.39** | **4.55 ± 0.35** | **+0.56**  |
> ### 6. Grammatical Errors
> We have addressed the mentioned issues. For the Lipschitz constraint, METRA uses L2 in the paper, though their code supports L1. Line 193 has also been revised for rigor.
>
> Thank you for your time and for recognizing our work! Due to space constraints, some explanations have been shortened — please feel free to reach out if any clarification is needed.

---

> > ### Comment · Reviewer_aHNd · 2025-04-01
> >
> > Thanks for the response. My main concerns have mostly been addressed.

---

> > > ### Author Response · Authors · 2025-04-07
> > >
> > > Thank you again for your recognition of our work! Your positive feedback means a lot to our team. We truly appreciate your time and thoughtful review. Wishing you all the best in your research and everyday life!

---

### Official Review · Reviewer_3FHL · 2025-03-10

**Overall Recommendation:** 2

**Summary:**

This paper propose a new unsupervised skill discovery method, which use regret to guide the skill sampling and skill policy learning. The regret is computed by the estimation error of value function in learning. Based on that, sampling strategies is not a parameters-constant distributions as previous methods. the paper claims the skill policy can prioritizes the exploration of under-converged skills during policy learning. This method improves the learning efficiency and increases skill diversity in simulation environment.

**Claims And Evidence:**

The claim of "incorporating regret awareness into skill discovery can enhance the learning efficiency" is supported by the experiment in Figure 3, with more higher Unique Coordinate measure during training.

**Essential References Not Discussed:**

Some reference I know are related to unsupervised skill discovery setting but seem not included in this paper:


[1] Explore, Discover and Learn: Unsupervised Discovery of State-Covering Skills.
Víctor Campos, Alexander Trott, Caiming Xiong, Richard Socher, Xavier Giro-i-Nieto, Jordi Torres

[2] Behavior Contrastive Learning for Unsupervised Skill Discovery
Rushuai Yang, Chenjia Bai, Hongyi Guo, Siyuan Li, Bin Zhao, Zhen Wang, Peng Liu, Xuelong Li

[3] Choreographer: Learning and Adapting Skills in Imagination
Pietro Mazzaglia, Tim Verbelen, Bart Dhoedt, Alexandre Lacoste, Sai Rajeswar

**Experimental Designs Or Analyses:**

The experiments conducted in the DM Control and D4RL environments are well-designed, with statistical significance reported, providing confidence in the results. The visualization experiment is also clear.

However, the experimental settings primarily focus on state-based environments, while the baselines compared like RSD and METRA, include more complex environments with pixel-based observations as input. This discrepancy makes it difficult to definitively conclude that the proposed method outperforms these baselines across a broader range of scenarios. Particularly, the reported sensitivity in skill generation (from Section 3.3) to hyperparameters raises concerns about the robustness of the method. This instability could limit its reliability and generalizability across a broader range of scenarios beyond the state-based environments tested.

**Methods And Evaluation Criteria:**

The regret-awared methods is well fit in unsupervised skill discovery problems.

**Other Comments Or Suggestions:**

Consider adding more experiments to strengthen applicability, e.g. pixel-based version of Kitchen as suggested by METRA^[1], or Unsupervised Reinforcement Learning Benchmark^[2].


[1] METRA: Scalable Unsupervised RL with Metric-Aware Abstraction. Seohong Park, Oleh Rybkin, Sergey Levine.

[2] URLB: Unsupervised Reinforcement Learning Benchmark. Michael Laskin, Denis Yarats, Hao Liu, Kimin Lee, Albert Zhan, Kevin Lu, Catherine Cang, Lerrel Pinto, Pieter Abbeel

**Other Strengths And Weaknesses:**

Strengths:
1. This paper is well-written and easy to read.
2. The regret-aware idea for skill discovery is novel.

Weaknesses:
1. Please see Experimental Designs Or Analyses above.

**Questions For Authors:**

1. How to caculate the Unique Coordinates in y-axis of Figure 3? it’s unclear whether the score (presumably represented on the y-axis) shows marginal improvement over the baselines or iterations. Could you clarify the calculation process and whether the trend indicates meaningful improvement? A detailed response would help me assess the significance of the reported results and could influence my evaluation of the method’s effectiveness.

**Relation To Broader Scientific Literature:**

The paper builds on skill discovery RL methods like compared baselines LSD, RSD, METRA. The "regret bonus" is a well-known concept in exploration RL setting.

**Theoretical Claims:**

This paper mainly focus on emperical results and no major issues of theroerm detected.

---

> ### Author Rebuttal · Authors · 2025-03-31
>
> We thank the reviewer for their feedback.
> ### Q1: Unique Coordinates Metric
> 1. **Origin of the Metric**:
> The metric is inspired by the *Policy State Coverage* measure introduced in the METRA paper (Section 5.3), where it was used to evaluate the **spatial coverage** of skill policies—referred to as x-y Coverage in the Maze environment. In our work, we rename it to *Unique Coordinates* to improve specificity, as noted in Section 4.1 (line 312).
> 2. **Definition**:
> The metric essentially counts the number of unique 2D coordinates visited by the agent during skill executions. We extract the x and y coordinates from states (hence the name Coordinates), then count the number of distinct positions (hence using Unique).
> 3. **Computation**:
> For each algorithm:
>    - Extract trajectory x-y states for each skill;
>    - Discretize the x-y values; (e.g., rounded to integers);
>    - Count the number of unique coordinate pairs across all rollouts.
> 4. **Significance**:
> This metric reflects how diversely the skills explore the state space. As supported in prior work, reaching distinct regions of the environment implies higher skill diversity(Figure 4).
>
> We will include this clarification in the revised version.
>
> ---
> ### Q2: Additional Experiments
> 1. **Kitchen**: It uses **pixel-based observations** and involves **manipulation** tasks, making it different from Ant-based settings.
> 2. **Motivation**: Following ([issue](https://github.com/seohongpark/METRA/issues/9#issuecomment-2565876012)), pixel-based tasks are indeed computationally demanding. We prioritized the Kitchen environment as it is both challenging and practically meaningful under time constraints.
> 3. **Experimental Design**: Following METRA’s setup, we report **the number of completed** tasks (7 in total) using  **mean ± std** over three seeds {0,2,4}.
> #### Performance @ Skill Dim = 24
> |**Model**| **Step = 100k**| **Step=200k** | **Step = 300k** | **Step = 400k** | **Δ@400k** |
> |:-:|:-:|:-:|:-:|:-:|:-:|
> |METRA|**2.72±0.19**|3.20±0.27|3.93±0.67|3.94±0.36||
> |Ours|2.41±0.02|**3.61±0.02**|**4.29±0.11**|**5.08±0.45**|+1.14|
> #### Performance @ Skill Dim = 32
> |**Model** | **Step = 100k** | **Step = 200k** | **Step = 300k** | **Step = 400k** | **Δ@400k** |
> |:-:|:-:|:-:|:-:|:-:|:-:|
> |METRA|3.21±0.71|3.66±0.69|3.83±0.90|3.99±0.94||
> |Ours|**3.51±0.23**|**3.91±0.43**|**4.34±0.39**|**4.55±0.35**|+0.56|
>
> Our method shows both **higher sample efficiency** and **stronger final performance**. At **dim = 32**, our method exhibits **faster** efficiency gains (e.g., Δ@300k = **+0.51**) compared to **dim = 24** (Δ@300k = **+0.36**), suggesting that our efficient approach benefits more from a larger latent space. This trend is even clearer when visualized in the learning curves.
>
> ---
> ### Q3: Hyperparameter Sensitivity and Stability
> 1. **Ablation**: We include ablations (Appendix E) for critical hyperparameters, such as the regret window size (`window`) and regularization weights (`alpha_1`, `alpha_2`).
> 2. **Tuning**: We argue that hyperparameter tuning in our method is **guided rather than purely search-based**, as they primarily influence skill diversity. For instance, the KL divergence term in Eq. (16) provides a direct and interpretable signal for adjusting `alpha_1`.
> 3. **Empirical Settings**:
>    - **Ant**/**Maze**: Used a fixed setting (`alpha_1=5`, `alpha_2=1`, `window=15`) across experiments.
>    - **Kitchen**: Used lighter settings (`alpha_1=1`, `alpha_2=0`, `window=8`) due to inherent diversity.
>
> While we also considered adaptive schemes (e.g., Lagrange multipliers), we avoided them to maintain a lightweight design.
>
> ---
> ### Q4: Related Work
> 1. **Explore, Discover and Learn: Unsupervised Discovery of State-Covering Skills**
>    This paper is cited in our Appendix B. It shares similar experimental setups with ours. However, we did not include it as a primary baseline because it focuses on simpler 2D navigation tasks and does not scale easily to high-dimensional settings.
> 2. **Behavior Contrastive Learning for Unsupervised Skill Discovery**
>    Thank you for the recommendation. Despite conceptual similarity, it lacks demonstration in high-dimensional environments. We also tried contrastive losses but observed unstable learning, potentially due to conflicting gradients in METRA. We acknowledge this as a promising direction.
> 3. **Choreographer: Learning and Adapting Skills in Imagination**
>    An insightful suggestion. The key differences are:
>    - It's a **model-based** RL approach, which increases sample efficiency but at a higher computational cost;
>    - It's **exploration-agnostic**, relying on separate exploration strategies (as noted in their Appendix A), whereas our method integrates exploration directly into the skill learning process.
>
> Although these works tackle skill discovery, we address different challenges. We will add these references to our revised Related Work section.
>
> We hope these resolve your concern!

---

> > ### Comment · Reviewer_3FHL · 2025-04-03
> >
> > Thank you for your response. The additional detailed information has addressed my concerns regarding certain definitions in this paper. The preliminary experiment comparing METRA with pixel-based observations shows promise; however, I believe the next version would benefit from more thorough experiments and in-depth analysis. Additionally, the paper should include further content and analysis on this topic. Based on these considerations, I have decided to maintain my score.

---

> > > ### Author Response · Authors · 2025-04-07
> > >
> > > Thank you very much for recognizing the promise of our work. We believe that the clarifications and additions made during the rebuttal process have significantly strengthened the paper, both in terms of clarity and technical contribution. We are confident that, with these improvements, the paper presents a meaningful and timely contribution to the community. Although you decided not to increase your score, we sincerely appreciate your thoughtful review and the time you dedicated to our submission.

---

### Official Review · Reviewer_42wC · 2025-03-12

**Overall Recommendation:** 3

**Summary:**

The paper presents Regret-aware Skill Discovery (RSD), a novel approach to unsupervised skill discovery in reinforcement learning. The authors conceptualize skill discovery as a min-max adversarial game between skill generation and policy learning. Their key insight is that skill discovery should be guided by policy strength convergence - focusing exploration on skills with weak, unconverged strength while reducing exploration for skills with already converged strength. To implement this, they use a regret measure to quantify policy strength convergence and employ a learnable skill generator within a population-based framework.

**Claims And Evidence:**

The main claims about RSD outperforming baselines in terms of efficiency and diversity are generally supported by the experimental results presented in the paper.
The authors show performance comparisons across multiple environments (Ant, Maze2d-large, Antmaze-medium, Antmaze-large), demonstrating improved coverage metrics and zero-shot performance.

However, the evidence would be stronger with clearer reporting of statistical significance - the paper lacks information about the number of independent seeds for experiments and doesn't provide error bars or uncertainty measurements for the reported results in Table 1.

**Essential References Not Discussed:**

To the best of my knowledge, all the relevant related works are cited/discussed in the paper.

**Experimental Designs Or Analyses:**

I checked the experimental design for comparing RSD against baseline methods.
While the experimental setup appears sound overall, there are several omissions:

1. The number of independent seeds run for each experiment is not specified
2. Statistical significance of results is not reported (no error bars in Figure 3 or uncertainty measures in Table 1)
3. There are inconsistencies in the reported timescales (Figure 3 shows results for 5e6 timesteps for Maze2d-large, while Figure 7 shows results for 1e7 timesteps for the same environment)
4. The AntMaze2D results mentioned in Appendix D appear to be missing.

**Methods And Evaluation Criteria:**

The proposed methods and evaluation criteria are reasonable for the problem of unsupervised skill discovery. The authors use appropriate metrics like state coverage (CoverCoords) to measure skill diversity and zero-shot performance to assess skill utility.

However, the CoverCoords metric could be more interpretable if reported as percentages of achievable coordinates on the map rather than absolute values.

The zero-shot experiment look a bit artifical to me: it shows improved results for RSD, but given that RSD achieves higher coverage (CoverCoords), it naturally has better chances of reaching goals.

**Other Comments Or Suggestions:**

Typos and presentation issues:

- Algorithm 1, line 10: should be $\pi_{\theta_1}$ instead of $\pi_{\theta_2}$
- Line 216: extra "the"
- Line 236, 2nd column: missing space before "Therefore"
- Figure 6: "Sapce" instead of "Space"
- Line 328: "DIYAN" instead of "DIAYN"
- Line 312, 2nd column: "the both" is awkward phrasing
- Line 380: "RSDlearns" should have a space
- Line 345, 2nd column: "Maeze2d-large" instead of "Maze2d-large"
- The colors in Figure 6 make it difficult to read

**Other Strengths And Weaknesses:**

Strengths:

- The paper is well-organized and clearly written.
- The algorithm is well-illustrated with helpful figures that explain the approach.
- The experiments demonstrate meaningful improvements over existing methods.
- The bounded representation learning approach appears to be a useful contribution.

Weaknesses:

- The authors claim RSD is a framework, but it's only implemented on top of METRA. To substantiate this claim, they should demonstrate applicability to other base algorithms like LSD.
- Ablation studies are relegated to the appendix (Appendix E) without being properly referenced in the main text.

**Questions For Authors:**

1. How many independent seeds were run for each experiment? What do the bold lines and shaded areas represent on the plots in Figure 3? What do the numbers in Table 1 represent (averages, medians)? What are the uncertainties in these measurements?
2. You claim RSD is a framework applicable beyond METRA. Have you tested applying it to other skill discovery algorithms such as LSD? If so, what were the results?
3. In Figure 6, how many timesteps does an "epoch" represent?
4. Why are the results in Figure 7 presented for 1e7 timesteps, whereas for the same environment in Figure 3, the results are presented for 5e6 timesteps?

**Relation To Broader Scientific Literature:**

The paper positions RSD appropriately within the unsupervised skill discovery literature, building upon temporal representation learning approaches like METRA.
The authors clearly acknowledge and compare to previous work on mutual information-based skill discovery (DIAYN, DADS, LSD, METRA).
They also explain how their approach differs by focusing on the relationship between skills and policy learning strength.
The regret-aware optimization bears some resemblance to concepts from Learning Process, which the authors acknowledge.

**Theoretical Claims:**

Not applicable (there are no theoretical claims)

---

> ### Author Rebuttal · Authors · 2025-03-27
>
> We sincerely thank the reviewer for the thoughtful and constructive feedback. Below we address each point in detail.
>
> ### **Q1. Statistical Reporting**
>
> We appreciate the reviewer’s emphasis on statistical rigor.
> All experiments were conducted using **five independent random seeds**: `{0, 2, 4, 8, 16}`.
>
> - In **Figure 3**, the **bold lines** indicate the **mean**, and the **shaded areas** represent the **standard deviation** across these runs.
> - In **Table 1**, we originally reported the **mean performance** only. In the revised version, we will **add uncertainty measures** to each entry. For example, in the *Maze2d-large* environment:
>
> ```
> RSD (ours):     FD↓ = 0.535 ± 0.152, AR↑ = 0.811 ± 0.046
> METRA-d:        FD↓ = 0.658 ± 0.154, AR↑ = 0.782 ± 0.043
> ```
> These results demonstrate that RSD achieves competitive performance with **comparable variance**, supporting the robustness of our method.
>
> ### **Q2. Applicability Beyond METRA**
>
> Thank you for the insightful question regarding generality. We certainlly plan to explore this framework with other models (for example in VLA) in future work.
>
> Our claim that RSD is a **general framework** refers to the **min-max optimization formulation in Equation 8**, which is **algorithm-agnostic** and, in principle, applicable to a wide range of **unsupervised skill discovery (USD)** methods.
>
> In this work, we chose to **implement RSD on top of METRA** rather than LSD. This is because **METRA**, an improved version of LSD, provides **key theoretical properties** (Equation 3) that we leverage. In Section 3.2 (Equations 2 and 13), we introduce **targeted modifications** to enhance performance in skill-asymmetric environments.
>
> ### **Q3. Epochs in Figure 6**
>
> We appreciate the reviewer’s comment regarding Figure 6.
>
> The relationship between **epochs and timesteps** is reflected in **Algorithm 1 (lines 2 and 5)**:
>
> ```
> timesteps = epoch × num_trajectories × max_length
> ```
>
> As described in **Appendix C.1**, we use:
>
> - `num_trajectories = 16`
> - `max_length = 300`
>
> So the approximate mapping is:
>
> ```
> 4000 epochs   → 2e7 timesteps
> 8000 epochs   → 4e7 timesteps
> 14000 epochs  → 7e7 timesteps
> 18000 epochs  → 9e7 timesteps
> ```
> In the revised version, we will **redraw Figure 6** with **explicit timestep annotations** and **improve color contrast**, as suggested, to improve readability.
>
> ### **Q4. Timestep Scales in Figures 3 and 7**
>
> Thank you for noting the difference in timestep scales.
>
> This design choice is **intentional**, as the two figures serve **different purposes**:
>
> - **Figure 3** highlights **early-phase learning efficiency** (shown up to **5e6** timesteps), with focus on the **0–2e6** range.
> - **Figure 7** illustrates **long-term convergence and stability**, plotted up to **1e7** timesteps.
>
> To avoid confusion, we will **explicitly clarify this** in the caption of Figure 7 in the final version.
>
> ### **Additional Comments**
>
> #### 1. Ablation Study Referencing
>
> We agree that the ablation study contains important insights. In the revised version, we will:
>
> - **Explicitly reference Appendix E in the main text**
> - **Summarize key findings** to highlight contributions of individual components
>
> #### 2. Missing AntMaze2D in Appendix D
>
> Thank you for catching this. **Appendix D contains visualized results** for experiments Table 1. We will:
>
> - **Clarify this reference in the main text**
> - Add **explanatory notes in Appendix D** to improve readability
>
> #### 3. Zero-shot Evaluation Concern
>
> We appreciate your thoughtful concern. The goal of our zero-shot experiment is to evaluate **practical utility**.
> Indeed, **RSD achieves superior zero-shot performance partly due to higher coverage** — which we see as a **desired property**.
> The improved zero-shot success rate is a **direct consequence of more effective skill discovery**, which we believe substantiates our central claim.
>
> ### **Typos and Presentation Issues**
>
> We thank the reviewer for the attention to detail. All identified **typos and presentation issues have been carefully corrected** in the revised version to improve clarity and presentation quality.
>
> Once again, we sincerely appreciate the reviewer’s time and valuable, constructive feedback. We believe it has significantly enhanced the clarity, rigor, and overall quality of our paper.

---

> > ### Comment · Reviewer_42wC · 2025-04-09
> >
> > Thank you very much for your response, my concerns have been addressed.
> > Given the paper's novel contribution and comprehensive baseline comparisons, I believe the paper should be accepted, and I've updated my score accordingly.
> >
> > Nonetheless, I agree with other reviewers that comparing RSD on more tasks would make paper stronger.

---

### Official Review · Reviewer_uCJY · 2025-03-17

**Overall Recommendation:** 1

**Summary:**

This paper is on unsupervised skill discovery within the context of Markov decision processes. It builds on a collection of earlier papers that aim to learn diverse and distinguishable behaviours, for example, DIAYN by Eysenbach et al., (ICLR 2019). The contribution of this paper is a new algorithm, which the authors call Regret-Aware Skill Discovery (RSD). Conceptually, their main idea is to prioritise the exploration of skills the authors call "under-converged" rather than indiscriminately exploring the whole skill space. The authors present an experimental evaluation of their approach.

**Claims And Evidence:**

I find that this paper is not clearly written. It is difficult to pinpoint exactly what the claims are. One claim seems to be that "skill discovery should be grounded by the policy strength" (page 1). I am not fully clear on what is meant by policy strength. In addition, given such a broad claim about skill discovery, the empirical evaluation should present evidence from a broad collection of skill discovery algorithms, in a broad range of problem domains; this is not the case in the paper.

A second claim is that the proposed method "significantly improves the learning efficiency and increases skill diversity in complex environments" and "achieves up to a 16% improvement in zero-shot evaluation compared to baselines". The support for these claims comes from a rather narrow experimental evaluation in terms of baseline algorithms considered and domains tested (details below) compared to the general practice in the literature (see, for example, METRA by Park et al, ICLR 2024).

**Essential References Not Discussed:**

Nothing particular.

**Experimental Designs Or Analyses:**

The primary issue with the experimental design is its narrow scope.

**Methods And Evaluation Criteria:**

The experiments are quite narrow in their scope. The authors can provide stronger support for their claims by evaluating their approach in a border range of domains and by comparing to a broader set of baseline algorithms. Currently, all experiments in the paper have been performed in the ant environment. In terms of algorithms, the baselines tested in the paper are only those that are very closely related to the proposed approach. If the claim is a new approach that "significantly improves the learning efficiency", a broader set of algorithms should be tested as baselines, for example, those that use intrinsic rewards.

**Other Comments Or Suggestions:**

Define Phi on line 131.

**Other Strengths And Weaknesses:**

The paper may have a useful contribution to make but requires a more clear statement of its claims and a more rigorous evaluation of those claims.

**Questions For Authors:**

Nothing particular.

**Relation To Broader Scientific Literature:**

The paper introduces a new approach within a particular line of research on unsupervised skill discovery. I am not able to judge its significance based on the relatively narrow analysis in the paper.

**Theoretical Claims:**

N/A

---

> ### Author Rebuttal · Authors · 2025-04-01
>
> ### Q1: Motivation of "Policy Strength"
> 1. **Definition:**
>
>     The term *policy strength*, derived from "the strength of the policy", refers to the **capability** of a policy to accomplish its designated objective—corresponding to **a specific skill** in our context.
> 2. **Why we use "policy strength":**
>
>     In this work, we adopt the term *policy strength* to concisely describe the **capability of the policy under skill-conditioned training**. More specifically, we are interested in how well the policy can be optimized given a particular skill, and whether it has **converged**, or **how far it is from convergence**. This perspective helps us assess whether a skill has been sufficiently explored or still requires further training attention.
> 3. **Motivation and necessity:**
>
>     Initially, we considered using phrases like *"the performance of the policy"*, but we were concerned that *performance* might be misinterpreted as referring to the cumulative reward. This can be misleading in unsupervised skill discovery, where intrinsic rewards are not always directly comparable between different skills. In contrast, *policy strength* focuses on the **optimization dynamics** and **convergence behavior** of the policy, thus avoiding ambiguity tied to reward metrics.
>
> We acknowledge the term may be unconventional and will revise the introduction to clearly define it.
>
> ---
> ### Q2: More Experimental Evaluation
> Thank you for encouraging broader validation.
> 1. **Choice of Maze Environment**
>
>     We extend METRA’s evaluation by including **Maze**, a more challenging setup for skill discovery [1]. As seen in *Figure 5*, METRA underperforms here due to the **skill-asymmetry issue**, which our method specifically targets.Additionally, we assess **zero-shot performance** on downstream tasks, following standard protocol.
> 2. **Evaluation on the Kitchen Environment**
>
>     In response to your concern, we additionally include experimental results on the **Kitchen**:
>     - **Kitchen Environment**:
>         Kitchen involves **pixel-based observations** and more complex robot-**manipulation tasks**, which are evaluated in METRA.
>     - **Why Kitchen**:
>         As noted in [this issue](https://github.com/seohongpark/METRA/issues/9#issuecomment-2565876012), pixel-based tasks are indeed **computationally demanding**. We chose Kitchen for its practical challenge and feasibility under limited time.
>     - **Experimental Setup**:
>         Following METRA’s setup, we measure the **number of completed tasks** (7 total) using **mean ± std** over three seeds `{0, 2, 4}`. We evaluate across different training steps and skill dimensions (`dim=24, 32`).
>         #### Performance @ Skill Dim = 24
>         |**Model**| **Step = 100k**| **Step=200k** | **Step = 300k** | **Step = 400k** | **Δ@400k** |
>         |:-:|:-:|:-:|:-:|:-:|:-:|
>         |METRA|**2.72±0.19**|3.20±0.27|3.93±0.67|3.94±0.36||
>         |Ours|2.41±0.02|**3.61±0.02**|**4.29±0.11**|**5.08±0.45**|+1.14|
>         #### Performance @ Skill Dim = 32
>         |**Model** | **Step = 100k** | **Step = 200k** | **Step = 300k** | **Step = 400k** | **Δ@400k** |
>         |:-:|:-:|:-:|:-:|:-:|:-:|
>         |METRA|3.21±0.71|3.66±0.69|3.83±0.90|3.99±0.94||
>         |Ours|**3.51±0.23**|**3.91±0.43**|**4.34±0.39**|**4.55±0.35**|+0.56|
>     - **Results Summary**:
>         Our method outperforms METRA in both **sample efficiency** and **final performance**. At **skill dim = 32**, our method exhibits greater gains with training progress (e.g., Δ@300k = +0.51 vs. +0.36 at dim=24). This trend is even clearer when visualized in the curves.
> ---
> ### Q3: Baseline Selection
> 1. **Baseline Selection**
>
>     Our method is built upon METRA’s framework. Our primary contribution lies in **enhancing the sampling efficiency** by incorporating ideas from **Prioritized Level Replay (PLR)**, enabling more effective skill discovery within the METRA setup.
> 2. **Why we do not compare against intrinsic reward-based methods**
>
>     As discussed in **Appendix A.1**, we chose not to include other intrinsic reward methods for the following reasons:
>     - **RND-based** methods improve exploration but are **not tailored for skill discovery**.
>     - **KNN-based** methods are limited in high-dimensional representation space.
>
>     Crucially, our focus is different: we optimize **skill distribution** during training using a regret-driven min-max formulation, unlike the uniform sampling in METRA and related work.
>
> We hope this explains our baseline decisions.
>
> ---
> ### Q4: Define $\phi$ on Line 131
> We will add the following definition to improve clarity:
> > A representation function $ \phi: \mathcal{S} \rightarrow \mathbb{R}^d $, which maps states into a skill-aligned latent space that shares the same dimensionality as the skill variable $ z \in Z $.
>
> ---
> [1] Campos, Víctor, et al. "Explore, discover and learn: Unsupervised discovery of state-covering skills." *International conference on machine learning*. PMLR, 2020.

---

### Decision · Program_Chairs · 2025-05-01

**Decision:**

Accept (poster)

**Comment:**

This paper frames unsupervised exploration as a min-max game, and find skills which maximize Regret, as this suggests that the policy has not yet fully converged -- building on the success of regret in other domains such as UED.

The use of these ideas in the unsupervised skill discovery is novel and a strong contribution, and was demonstrated effective in a number of experimental domains against competitive baselines. We would encourage the authors to follow through on the improvements to the related work, writing clarity and exposition of the statistical significance of their results as detailed throughout the rebuttal period discussion.